# P-LoRA: Posterior Knowledge Enables Training-Free Fusion of Subject and Style LoRAs

## Abstract

Recent studies have explored the combination of multiple LoRAs to simultaneously generate learned subjects and styles. However, most existing approaches fuse LoRA weights directly based on their statistical properties, which deviates from the original intent of LoRA, namely learning additional features to adapt to diverse functions. To address this limitation, we introduce **P-LoRA**, a new training-free fusion paradigm that leverages posterior knowledge from fine-tuned features, fundamentally shifting the fusion process from weight-level heuristics to representation-conditional decisions. Specifically, at each LoRA-applied layer, we compute the KL divergence between the original features and the features generated by subject and style LoRAs, respectively, to adaptively select the most appropriate weights for fusion. Furthermore, objective metrics such as CLIP and DINO scores, which reflect alignment and semantic consistency, are employed as posterior knowledge to dynamically adjust denoised embeddings during the generation process. By incorporating posterior knowledge into the fusion pipeline, P-LoRA effectively preserves the most representative subject and style characteristics without requiring retraining. Extensive experiments across diverse subject-style combinations demonstrate that P-LoRA consistently outperforms existing methods, achieving superior results both qualitatively and quantitatively.

## 1 Introduction

Diffusion models have demonstrated remarkable performance across a wide range of generative tasks Chen et al. (2024); Jiang et al. (2024b); Gupta et al. (2024); Xing et al. (2024b); Zheng et al. (2023); Ma et al. (2023). Among these, personalized image generation Ruiz et al. (2023); Sohn et al. (2023) has garnered increasing attention, as it requires the model to synthesize high-quality images that reflect user-specified content or style. Here, content refers to the semantic structure and subject identity, while style captures visual properties such as color, texture, and patterns. Although substantial progress has been made in generating images conditioned on either content or style alone, producing images that faithfully integrate both a specific subject and a specific style remains a challenging and unsolved problem.

Recently, Low-Rank Adaptation (LoRA) Hu et al. (2022) has emerged as a popular and versatile technique for parameter-efficient fine-tuning, making it particularly appealing for personalized generation tasks. Leveraging the modular nature of LoRA, recent studies have explored the fusion of independently fine-tuned LoRAs to jointly generate specific subjects in specific styles. For example, ZipLoRA Shah et al. (2024) proposes leveraging coefficient vectors to merge content and style LoRAs in each LoRA-applied layer. Differently, B-LoRA Frenkel et al. (2024) investigates the impact of diverse LoRA layers and finds that modifying two distinct LoRA layers can effectively control the content and style of generated images. Furthermore, focusing on the intrinsic characteristics of LoRA weights, K-LoRA Ouyang et al. (2025) selects LoRAs in each layer by comparing the Top-K elements of the weights. While these methods have demonstrated promising performance in LoRA fusion, as shown in Figure 1 (a), their core strategies remain grounded in statistical properties of LoRA weights, which diverge from the original intent of LoRA—learning additional features to adapt to diverse functions. This divergence suggests that the fine-tuned features themselves, rather than the LoRA weights alone, are the true key to effective fusion.

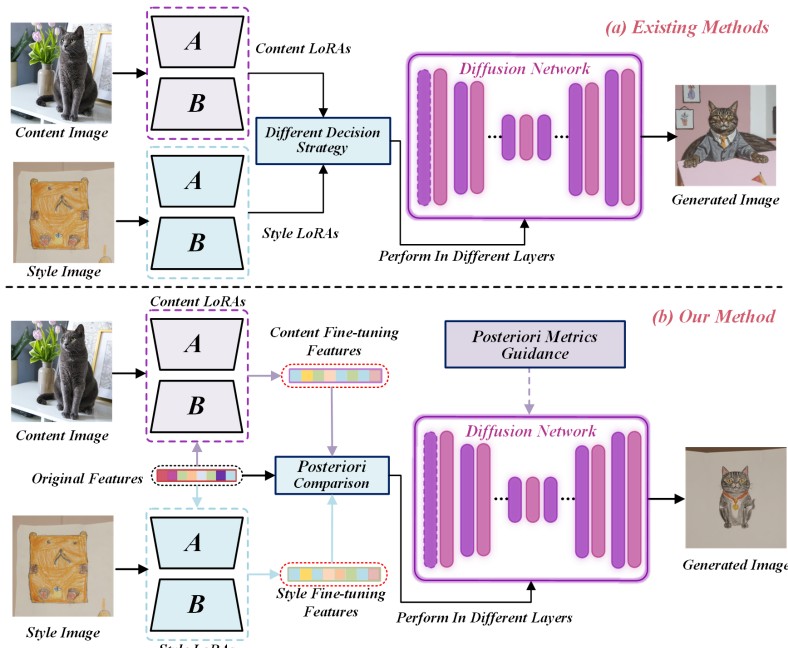

Figure 1: (a) existing methods directly rely on properties of LoRA weights to achieve fusion. (b) Our method leverages posterior knowledge from fine-tuned features and objective metrics to enable training-free fusion of subject and style LoRAs.

Inspired by this, as shown in Figure 1 (b), we introduce a novel training-free fusion paradigm based on posterior knowledge from fine-tuned features and metrics, which fundamentally shifts the fusion process from weight-level heuristics to representation-aware decisions. In particular, the previous method Ouyang et al. (2025) argues that the absolute values of LoRA weights indicate their importance in the diffusion process. By contrast, we propose that the feature changes induced by LoRAs serve as a more direct and key indicator of their impact. In each LoRA-applied layer, we compute the fine-tuned features from both the style LoRAs and the content LoRAs, respectively. To better quantify the extent of feature change, we leverage the Kullback-Leibler (KL) divergence between the fine-tuned features and the original features, determining which LoRA is more suitable for each layer based on the magnitude of distributional change. In this way, we adaptively retain the most significant features in each LoRA-applied layer, thereby preserving the most representative content and style information.

Moreover, objective metrics such as CLIP Radford et al. (2021) and DINO Caron et al. (2021) scores can effectively assess the quality of LoRAs fusion. We therefore adopt these scores as posterior knowledge to guide the generation process by dynamically adjusting the denoised embeddings. Specifically, to obtain reference images for the metrics, the content and style LoRAs are used to generate corresponding reference images for content and style, respectively. During the denoising process, we evaluate the quality of intermediate generated images using these metrics and apply gradient-based guidance to steer the denoised embeddings toward higher-quality results. In this manner, objective metrics serve as posterior knowledge to provide beneficial guidance throughout the denoising process, ultimately enabling the generation of high-quality images that effectively integrate desired subject and style.

Through leveraging posterior knowledge from fine-tuned features and objective metrics in the training-free manner, P-LoRA achieves effective integration of subject and style LoRAs, enabling high-quality image generation that preserves both subject fidelity and stylistic accuracy in a training-free manner. The main contributions of this work are summarized as follows:

- P-LoRA introduces a novel training-free LoRA fusion paradigm that fundamentally shifts the fusion process from weight-level heuristics to representation-conditional decisions by leveraging posterior knowledge from fine-tuned features and objective metrics.

- A KL divergence-based adaptive fusion strategy dynamically selects the most suitable LoRA weights to preserve subject and style. Objective metrics such as CLIP and DINO scores

are incorporated as posterior knowledge to provide gradient-based guidance during the denoising process, further enhancing semantic alignment and visual quality.

- P-LoRA is training-free, user-friendly, and demonstrates promising generalization ability, achieving superior performance across diverse subject-style combinations on multiple benchmarks without requiring retraining or additional supervision.

## 2 RELATED WORK

**Diffusion Models for Custom Generation.** With the rapid development of diffusion models He et al. (2024); Cao et al. (2024); He et al. (2025); Ho et al. (2022); Zhang et al. (2023a), many researchers have introduced diverse approaches to fine-tuning large-scale diffusion models for custom generation, which aims to produce images of user-specified subjects or styles based on language descriptions. For instance, Textual Inversion Gal et al. (2022) focuses on optimizing a single word embedding to capture unique and varied concepts. DreamBooth Ruiz et al. (2023) designs text prompts containing a unique identifier to more effectively generate images with the desired subjects. CustomDiffusion Kumari et al. (2023) fine-tunes the cross-attention layers within the diffusion model to learn multiple concepts simultaneously. Additionally, some methods Avrahami et al. (2023); Shi et al. (2024); Xie et al. (2023); Xiao et al. (2024) achieve custom generation without additional training, yet these typically target specific single tasks. Recently, parameter-efficient fine-tuning techniques such as LoRA Hu et al. (2022) and StyleDrop Sohn et al. (2023) have gained popularity due to their ability to fine-tune models with low-rank adaptations, making them especially attractive for custom generation.

**LoRAs combination for image generation.** Since the rise in popularity of LoRA applications Zhang et al. (2023b); Zhou et al. (2024); Zi et al. (2023), many studies on LoRA combinations have been proposed. Some methods Dong et al. (2024); Gu et al. (2023); Jiang et al. (2024a); Xing et al. (2024a) focus on fusing multiple object LoRAs, enabling diffusion models to generate various new concepts and replace these objects through masking strategies. Meanwhile, several advanced methods address content-style LoRA fusion. For instance, Mixture-of-Subspaces Wu et al. (2024) designs learnable mixer weights to fuse various LoRAs; ZipLoRA Shah et al. (2024) leverages merge vectors across varying layers to linearly combine subject and style LoRAs; B-LoRA Frenkel et al. (2024) investigates the impact of different LoRA layers and finds modifying two distinct layers can effectively control the content and style of generated images; and K-LoRA Ouyang et al. (2025) selects the appropriate LoRAs in each layer by comparing top-K elements of different LoRA weights. Although these methods have shown promising performance, they directly rely on the properties of LoRA weights. Unlike these approaches, we argue that since the original intent of LoRA is to learn additional features to adapt to diverse tasks, fusion methods based on fine-tuned features could be more effective. To this end, we introduce a novel training-free LoRA fusion paradigm that fundamentally shifts the fusion process from weight-level heuristics to representation-conditional decisions.

## 3 PRELIMINARIES

**Diffusion Models.** Diffusion models Saharia et al. (2022); Kazerouni et al. (2022); Chen et al. (2023); Amit et al. (2021) have demonstrated impressive performance across various generative tasks. They mainly consist of a forward noise addition process and a reverse denoising process. During the forward process, the original image is progressively transformed into Gaussian noise through incremental noise addition. In the reverse process, conditioned on paired text prompts, the diffusion network, typically a U-Net, gradually denoises the noisy input step-by-step, starting from randomly sampled pure noise. At inference time, the trained diffusion network achieves text-to-image generation based on the given textual input.

**LoRA.** Low-Rank Adaptation (LoRA Hu et al. (2022)) is a lightweight fine-tuning technique originally developed for large language and diffusion models. Rather than updating the full parameter matrix $W_0$, LoRA exploits the observation that the update $\Delta W \in \mathbb{R}^{m \times n}$ often lies in a low-dimensional subspace. Concretely, one factorizes $\Delta W = BA$ with $B \in \mathbb{R}^{m \times r}$ and $A \in \mathbb{R}^{r \times n}$ for $r \ll \min(m, n)$, and only $A, B$ are learned while $W_0$ remains fixed. The tuned model thus uses weights $W_0 + BA$. In our setting, let $D$ be a diffusion model with base weights $W_0$. To capture a new concept, we train a LoRA pair $\Delta W_x$ so that the adapted model is $D_x = W_0 + \Delta W_x$.

**Posterior Knowledge.** The primary goal of LoRA is to learn fine-grained feature adjustments that guide a diffusion model toward specific behaviors. To leverage this capability, we first generate the modified features produced by both the content and style LoRAs. While prior work Ouyang et al.

(2025) uses the magnitudes of LoRA weight updates as a proxy for their influence, we argue that the actual change in feature distributions provides a more direct and interpretable measure of impact. Guided by this *posterior knowledge*, we compare the original features with those modified by each LoRA, and use the resulting divergence to select the most informative contributions for fusion.

In addition, objective metrics such as CLIP and DINO scores offer an effective way to assess the quality of LoRA fusion. Higher scores indicate better alignment with the intended semantics or style, and thus can serve as *posterior knowledge* to guide the diffusion network. The guidance score $R$ based on the CLIP metric is computed as:

$$R(\hat{x}_0) = 1 - S_{\text{CLIP}}(x_{\text{ref}}, \hat{x}_0), \tag{1}$$

where $S_{\text{CLIP}}$ computes the CLIP similarity score, $x_{\text{ref}}$ is the reference image, and $\hat{x}_0$ is the predicted original image at step $t$.

The guiding scores act as residuals that can be treated as virtually observed values Kaltenbach & Koutsourelakis (2020) with $\hat{R} = 0$ and virtual likelihood:

$$p(\hat{R} = 0 \mid x_t) = \mathcal{N}(0 \mid R(x_0), \sigma_r^2 I), \tag{2}$$

where $x_t$ is the intermediate result at timestep $t$, and $\sigma_r$ is a predefined constant controlling the enforcement strength of the virtual observation. Although residuals are minimized during the generation process, they are not guaranteed to reach zero. To incorporate this guidance into the diffusion process, we apply Bayesian rule:

$$p(x_t \mid \hat{R} = 0) = \frac{p(x_t)\, p(\hat{R} = 0 \mid x_t)}{p(\hat{R} = 0)}. \tag{3}$$

Taking the gradient of the log-likelihood with respect to $x_t$, we obtain:

$$\nabla_{x_t} \log p(x_t \mid \hat{R} = 0) = \nabla_{x_t} \log p(x_t) + \nabla_{x_t} \log p(\hat{R} = 0 \mid x_t), \tag{4}$$

where the first term is the standard score function predicted by the diffusion model Song et al. (2020). For the second term, substituting Eq. 2 yields:

$$\nabla_{x_t} \log p(\hat{R} = 0 \mid x_t) = -\frac{1}{\sigma_r^2} \nabla_{x_t} \|R(x_0)\|_2^2 \approx -\frac{1}{\sigma_r^2} \nabla_{x_t} \|R(\hat{x}_0)\|_2^2, \tag{5}$$

where we approximate $x_0$ with $\hat{x}_0$. In practice, because guidance scores range between $[0, 1]$, we simplify as:

$$\nabla_{x_t} \log p(\hat{R} = 0 \mid x_t) \propto -\nabla_{x_t} R(\hat{x}_0). \tag{6}$$

Thus, to implement the guidance, the denoising step is modified as:

$$x_{t-1} = x_{t-1}^{\text{ori}} - m \nabla_{x_t} R(\hat{x}_0), \tag{7}$$

where $x_{t-1}^{\text{ori}}$ is the original output of the $t$-step denoising process, $\hat{x}_0$ is the predicted original image at step $t$, and $m$ is a predefined scaling factor.

## 4 METHOD

As illustrated in Figure 2, P-LoRA guides a base diffusion model $D$ to generate a specified subject in a specified style by leveraging both content and style LoRAs. Let the base model $D$ consist of pre-trained weights $W_0^i$ at layer $i$. Applying the LoRA $L_x$ with weight updates $\{\Delta W_x^i\}$ yields the adapted model:

$$D_{L_x} = D \odot L_x = W_0 + \Delta W_x. \tag{8}$$

In the experimental setting, we are given the content LoRA weights $L_c\{\Delta W_c^i\}$, the style LoRA weights $L_s\{\Delta W_s^i\}$, and the base model $D$. By combining posterior feature knowledge through comparisons of original and fine-tuned feature distributions along with posterior metric knowledge guided by metrics scores during the denoising process, P-LoRA enables the diffusion model to effectively render the target subject in the desired reference style without any additional training.

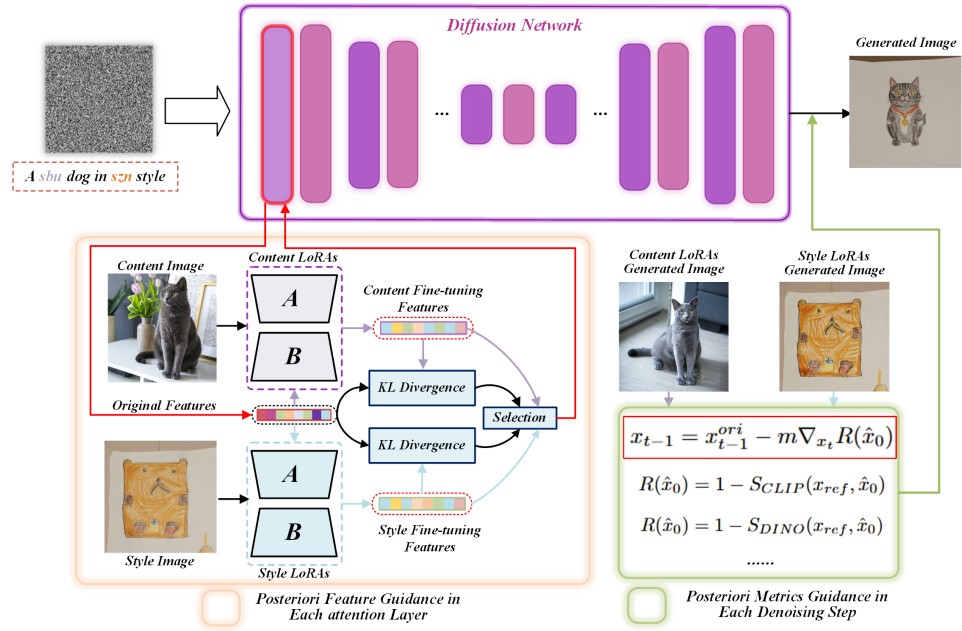

Figure 2: Overview of our proposed method. By incorporating posterior feature knowledge through distributions comparison and incorporating posterior metrics knowledge via score guidance, P-LoRA enables effective generation of the given object in a desired reference style.

### 4.1 POSTERIOR FEATURE KNOWLEDGE

Prior work has used the absolute values of LoRA weight updates as a proxy for their importance in diffusion models Ouyang et al. (2025). However, the core function of LoRA is to induce feature-level adjustments rather than merely altering weight magnitudes. Inspired by this notion of *posterior feature knowledge*, we explicitly examine how content and style LoRA weight updates, $\Delta W_c^i$ and $\Delta W_s^i$, affect the base network's $i$-th layer. Specifically, we apply these updates to the base weights $W_0^i$ and compute the corresponding fine-tuned feature maps:

$$\hat{F}_c^{\,i+1} = \left(W_0^i + \Delta W_c^i\right) F_i, \tag{9}$$

$$\hat{F}_s^{\,i+1} = \left(W_0^i + \Delta W_s^i\right) F_i, \tag{10}$$

where $F_i$ represents the original features at layer $i$.

To quantify the impact of these modifications, we compute the KL divergence between each fine-tuned feature distribution and the original feature distribution $F_{i+1}$:

$$d_c^i = \text{KL}\left(\hat{F}_c^{\,i+1} \parallel F_{i+1}\right), \tag{11}$$

$$d_s^i = \text{KL}\left(\hat{F}_s^{\,i+1} \parallel F_{i+1}\right). \tag{12}$$

We then compare $d_c^i$ and $d_s^i$ to determine which adjustment induces a more significant feature change:

$$F^{i+1} = \begin{cases} \hat{F}_c^{\,i+1}, & \text{if } d_c^i \geq d_s^i, \\ \hat{F}_s^{\,i+1}, & \text{otherwise.} \end{cases} \tag{13}$$

By performing this selection at each layer, we retain the most impactful content or style information, enabling an effective, training-free fusion of subject and style LoRAs.

In contrast to weight-based fusion strategies, which are static and input-agnostic, our feature-based approach dynamically adapts to the input: as the prompt changes, so do the feature distributions and thus the fusion decisions. This input-conditional mechanism allows P-LoRA to flexibly handle generation tasks with diverse and evolving requirements.

### 4.2 POSTERIOR METRICS KNOWLEDGE

As discussed in the preliminaries, objective metrics such as CLIP and DINO scores effectively assess the quality of LoRA fusion, where higher scores indicate better alignment with the desired content

or style. To obtain the guidance scores, we first leverage the content LoRAs $\Delta W_c$ and style LoRAs $\Delta W_s$ with content descriptions $l_c$ and style descriptions $l_s$ to generate the new reference content image $I_c^{\mathrm{ref}}$ and reference style image $I_s^{\mathrm{ref}}$:

$$I_c^{\mathrm{ref}} = (W_0 + \Delta W_c)[l_c] \tag{14}$$

$$I_s^{\mathrm{ref}} = (W_0 + \Delta W_s)[l_s] \tag{15}$$

With the original output $x_{t-1}^{\mathrm{ori}}$ at the $t$-th denoising step and the predicted original image $\hat{x}_0$ at step $t$, we compute the CLIP and DINO scores Shah et al. (2024) to evaluate the prediction performance. The scores are obtained by extracting embeddings from the generated and reference images and computing cosine similarity:

$$S_{\mathrm{CLIP}}^{\mathrm{content}} = \mathrm{Sim}_{\cos}\big(E_{\mathrm{CLIP}}(I_c^{\mathrm{ref}}), E_{\mathrm{CLIP}}(\hat{x}_0)\big) \tag{16}$$

$$S_{\mathrm{CLIP}}^{\mathrm{style}} = \mathrm{Sim}_{\cos}\big(E_{\mathrm{CLIP}}(I_s^{\mathrm{ref}}), E_{\mathrm{CLIP}}(\hat{x}_0)\big) \tag{17}$$

$$S_{\mathrm{DINO}}^{\mathrm{style}} = \mathrm{Sim}_{\cos}\big(E_{\mathrm{DINO}}(I_s^{\mathrm{ref}}), E_{\mathrm{DINO}}(\hat{x}_0)\big) \tag{18}$$

where $E_{\mathrm{CLIP}}$ and $E_{\mathrm{DINO}}$ denote the image encoders of CLIP Radford et al. (2021) and DINO Caron et al. (2021), respectively. To compute the final guidance score, we evenly weight the three metrics:

$$R(\hat{x}_0) = 1 - \frac{S_{\mathrm{CLIP}}^{\mathrm{content}} + S_{\mathrm{CLIP}}^{\mathrm{style}} + S_{\mathrm{DINO}}^{\mathrm{style}}}{3} \tag{19}$$

With guidance scores and empirically setting scaling factor $m = 10$, we guide the diffusion step as:

$$x_{t-1} = x_{t-1}^{\mathrm{ori}} - m\nabla_{x_t} R(\hat{x}_0) \tag{20}$$

By incorporating posterior metrics knowledge into the training-free denoising step, the generated images are dynamically steered toward higher quality and exhibit the desired subject and style, thanks to continuous supervision from the generated reference images.

In summary, P-LoRA leverages posterior feature knowledge to dynamically fuse content and style LoRAs at the feature level and integrates posterior metrics knowledge to guide denoising process with continuous objective feedback. This dual perspective ensures a training-free yet highly adaptive fusion mechanism. In the following section, we present extensive experiments to validate the effectiveness of P-LoRA across various generation tasks and compare it with existing advanced methods.

## 5 EXPERIMENTS

We evaluate the proposed **P-LoRA** approach under the experimental setup established by previous methods including K-LoRA, ZipLoRA, B-LoRA. Specifically, we apply P-LoRA to both the Stable Diffusion XL v1.0 base model and the FLUX model.

**Datasets.** For training the local LoRAs, we follow the convention of previous works Ouyang et al. (2025); Shah et al. (2024). To train content LoRAs, we select diverse image sets from the DreamBooth dataset Ruiz et al. (2023), where each instance is represented by 4–5 images. For style LoRAs, we adopt the dataset introduced by the StyleDrop authors Sohn et al. (2023), which includes a wide variety of stylistic exemplars spanning classical art to modern creative styles. Each style LoRA is trained using a single reference image.

**Implementation Details.** To obtain local LoRAs, we adopt the K-LoRA Ouyang et al. (2025) strategy to fine-tune the SDXL v1.0 base model using a low-rank adaptation with rank set to 64. The LoRA weights—both style and content—are optimized using the Adam optimizer over 1000 steps with a batch size of 1 and a learning rate of 5e-5. For the FLUX model, we utilize publicly available, well-trained community LoRA weights obtained from HuggingFace. Corresponding experimental results on FLUX are provided in the following appendix A.

### 5.1 RESULTS

#### 5.1.1 QUANTITATIVE COMPARISONS

To objectively evaluate the performance of our training-free P-LoRA method, we adopt commonly used metrics from prior works, including Style Similarity, CLIP Score, and DINO Score, to assess the quality of the generated images. Following previous methods, we randomly selected 30 unique content–style pairs, each of which consists of 10 images to perform quantitative comparisons.

| Method | Style Sim ↑ | CLIP Score ↑ | DINO Score ↑ |
|---|---|---|---|
| **Direct** | 48.9% | 66.6% | 43.0% |
| **B-LoRA** Frenkel et al. (2024) | 58.0% | 63.8% | 30.6% |
| **ZipLoRA** Shah et al. (2024) | 60.4% | 64.4% | 35.7% |
| **K-LoRA** Ouyang et al. (2025) | 58.7% | 69.4% | 46.9% |
| **P-LoRA (ours)** | 63.0% | 78.5% | 43.3% |

Table 1: Comparison of alignment results. Direct denotes direct arithmetic merging.

| Method | User Preference | GPT-4o Feedback | Qwen2.5-VL Feedback |
|---|---|---|---|
| **ZipLoRA** Shah et al. (2024) | 13.80% | 20.13% | 3.40% |
| **B-LoRA** Frenkel et al. (2024) | 21.89% | 11.67% | 9.11% |
| **K-LoRA** Ouyang et al. (2025) | 11.11% | 12.56% | 21.82% |
| **P-LoRA(ours)** | 53.20% | 55.64% | 65.67% |

Table 2: The Performance Comparison of user study results, GPT-4o and Qwen2.5-VL feedback.

Specifically, CLIP Radford et al. (2021) is employed to evaluate both the style alignment (*Style Sim*) and content preservation (*CLIP Score*), while DINO Zhang et al. (2022) is used to measure content consistency via *DINO Score*. Table 1 presents a detailed comparison between P-LoRA and existing state-of-the-art methods. Our method achieves the best performance in both *Style Sim* (63.0%) and *CLIP Score* (78.5%). Notably, P-LoRA provides a substantial improvement of 9.1% in *CLIP Score* compared to the strongest baseline. Although P-LoRA does not achieve the top performance in *DINO Score*, it still ranks second, demonstrating a strong overall balance between style and content fidelity. These results validate the effectiveness of our posterior-guided fusion method. As a supplementary verification, we report further evaluations with extra metrics in Appendix B, which also demonstrate the effectiveness of our proposed method.

### 5.1.2 USER STUDY AND MLLM-BASED EVALUATIONS

To further assess the perceptual quality of generated images beyond conventional metrics, we conduct a comprehensive user study and multimodal large language model (MLLM)-based evaluations. As shown in Table 2, we collect human preferences and automatic feedback from two strong MLLMs, GPT-4o OpenAI & Microsoft (2024) and Qwen2.5-VL Bai et al. (2025). In the user study, participants were asked to choose their preferred images from outputs of four competing methods. Our proposed P-LoRA is overwhelmingly favored, receiving 53.20% of total votes, outperforming all baselines. Similarly, in LLM-based evaluations, P-LoRA is consistently ranked highest, achieving 55.64% preference by GPT-4o and an even more substantial 65.67% by Qwen2.5-VL. These results not only confirm the good quality of generated images in terms of human preference but also highlight P-LoRA's effectiveness in producing stylistically and semantically coherent outputs that align well with multi-modal models. Setup details are provided in the following appendix C.

### 5.1.3 QUALITATIVE COMPARISONS

To visually assess the performance of different LoRA fusion methods, we present qualitative comparisons in Figure 3. Overall, our training-free P-LoRA demonstrates superior visual quality, effectively preserving both content and style information. In contrast, most existing methods tend to retain content reasonably well but struggle to capture the target style faithfully. For example, in the second row and third column, B-LoRA correctly identifies the 'dog' content but incorrectly applies a pink color inconsistent with the reference style. A similar issue is observed with K-LoRA in the fourth row and third column. Moreover, in the second-to-last row, K-LoRA produces a 'cat' whose head and body exhibit inconsistent styles, indicating a failure in achieving global style coherence. In the eighth row, K-LoRA also fails to preserve the oil painting style entirely. Interestingly, Zip-LoRA, despite being the only method with learnable parameters, performs relatively worse. It often fails to capture the desired style (e.g., the 'dog' in the fourth column) or generates semantically inaccurate content. These qualitative results further support the effectiveness of our posterior-guided fusion strategy, which consistently delivers visually coherent outputs without the need for additional training.

### 5.2 ABLATION STUDIES

To thoroughly analyze the contributions of each component in our proposed training-free P-LoRA pipeline, we conduct extensive ablation studies across some aspects: (1) the effect of Posterior

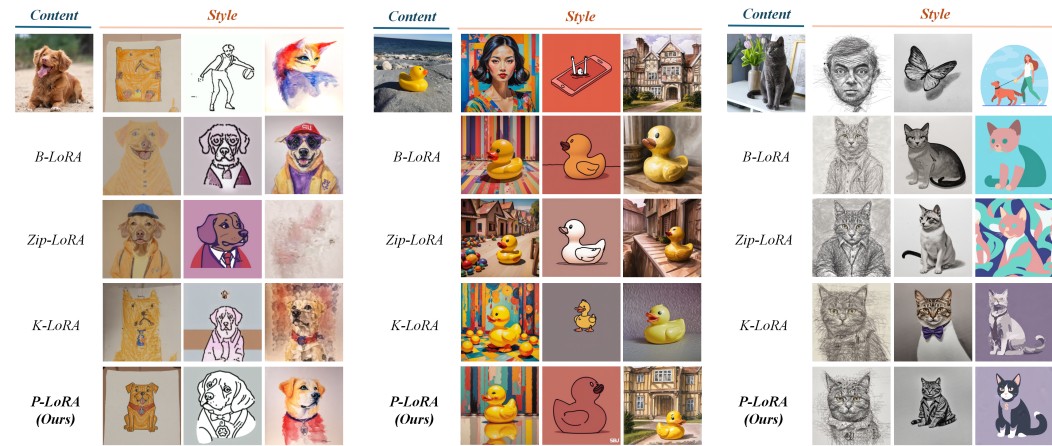

Figure 3: **Qualitative comparisons.** We present images generated by P-LoRA and the compared advanced generation methods. Through incorporating posterior knowledge, our method effectively enables training-free fusion of subject and style LoRAs.

| Method | Style Sim ↑ | CLIP Score ↑ | DINO Score ↑ |
|---|---|---|---|
| **Baseline** | 60.3% | 75.6% | 40.1% |
| **Only with PFK** | 59.5% | 75.9% | 43.7% |
| **Only with PMK** | 66.1% | 78.9% | 40.7% |
| **P-LoRA(PFK+PMK)** | 64.0% | 79.1% | 43.4% |

Table 3: Ablation study of different components, including Posterior Feature Knowledge (PFK) and Posterior Metrics Knowledge (PMK).

| Divergence | Style Sim ↑ | CLIP Score ↑ | DINO Score ↑ |
|---|---|---|---|
| **KL** | 59.5% | 75.9% | 43.7% |
| **JS** | 59.3% | 75.3% | 43.1% |
| **Cosine Similarity** | 58.9% | 75.4% | 43.3% |
| **Dot Product** | 58.4% | 74.9% | 43.8% |

Table 4: Ablation study of selection criteria in posterior guidance, including Kullback-Leibler (KL), Jensen-Shannon (JS) divergence, cosine similarity, and dot product.

| Scaling Factor $m$ | Style Sim ↑ | CLIP Score ↑ | DINO Score ↑ |
|---|---|---|---|
| **1** | 62.1% | 72.8% | 32.4% |
| **5** | 64.8% | 74.1% | 32.3% |
| **10** | 64.0% | 79.1% | 43.4% |
| **20** | 66.8% | 76.8% | 35.1% |

Table 5: Ablation study of scaling factor $m$ for posterior metrics knowledge.

Feature Knowledge (PFK) and Posterior Metrics Knowledge (PMK), (2) the selection criteria in posterior guidance., and (3) the impact of scaling the posterior-guided weighting. For each setting, we randomly sample 25 object-style combinations to ensure robustness.

**Effect of Posterior Feature and Metrics Knowledge.** Table 3 presents the performance of varying configurations. Incorporating PFK alone improves *DINO Score* by 3.6% over the baseline, while PMK boosts *Style Slim*(66.1%) and *CLIP Score*(78.9%). When using both components, the P-LoRA model achieves the best overall results, especially in *CLIP Score* (79.1%) and *DINO Score* (43.4%), demonstrating the complementary nature of latent feature knowledge and metric-aware refinement.

**Selection Criteria in Posterior Guidance.** To investigate the role of the selection function in posterior feature guidance, we compare Kullback-Leibler (KL), Jensen-Shannon (JS) divergence, cosine similarity, and dot product in Table 4. Clearly, KL and JS divergence brings more pleasant performance. While they both achieve comparable style and CLIP scores, KL slightly outperforms JS in *DINO Score* (43.7% vs. 43.1%), which aligns with the intuition that KL is more sensitive to asymmetrical discrepancies in content distributions, making it more suitable for our setup.

**Effect of Scaling Factor.** We also explore the sensitivity of P-LoRA to the scaling factor that balances the influence of posterior metric knowledge, as shown in Table 5. Setting this factor too low (e.g., 1)

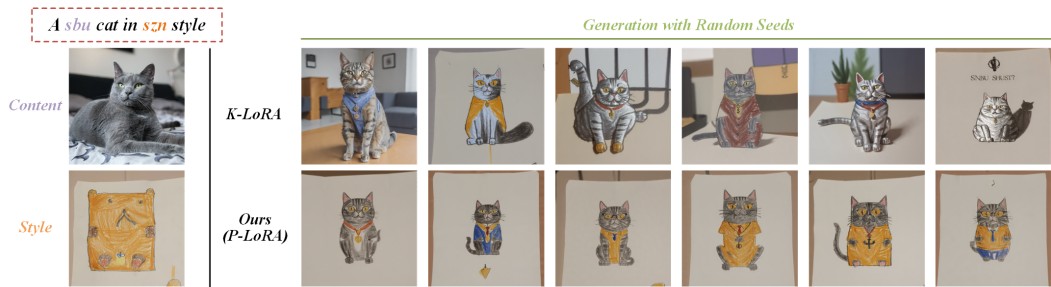

Figure 4: **Robustness Analysis.** We present images generated by P-LoRA and K-LoRA with random seeds to analyze the robustness.

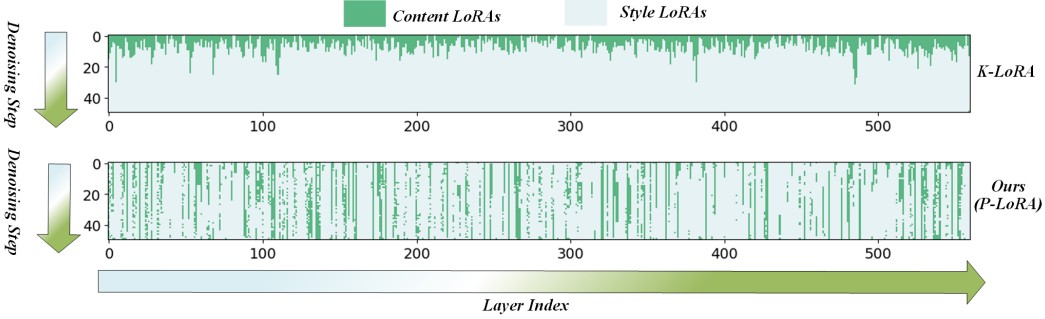

Figure 5: **LoRA Selection During the Generation Process.** The vertical axis represents 50 diffusion steps, while the horizontal axis denotes varying LoRA layers. Dark green indicates the selection of subject LoRA, and light blue indicates the selection of style LoRA.

leads to suboptimal *CLIP* and *DINO Scores*, while extremely high scaling (e.g., 20) causes instability in *DINO Score*. A moderate value of 10 yields the best trade-off, with a strong boost in both *CLIP Score* (79.1%) and *DINO Score* (43.4%), highlighting the importance of proper posterior calibration.

**Robustness Analysis.** To further evaluate the robustness of our proposed P-LoRA framework, we visualize generated results under different random seeds and compare them with K-LoRA in Figure 4. While K-LoRA exhibits significant variability in scene layouts and fails to consistently preserve the target style, P-LoRA maintains both the semantic content and stylistic attributes across different sampling conditions. This stability under stochastic perturbations highlights the effectiveness of posterior-guided modulation in enforcing coherent and reliable image generation.

**Visualization of Posterior Knowledge Selection.** Figure 5 illustrates the dynamic posterior selection mechanism employed by P-LoRA. Unlike K-LoRA, which relies solely on static weight elements for selection, our method performs input-conditional selection, dynamically choosing the more relevant LoRA weights based on input-dependent features. This adaptive strategy enables P-LoRA to better align with the input semantics, thereby facilitating more effective and coherent style-content fusion. The superior performance observed in previous quantitative and qualitative comparisons further validates the advantages of this training-free, input-adaptive selection scheme.

# 6 CONCLUSION

In this paper, we propose P-LoRA, a novel training-free LoRA fusion paradigm that fundamentally shifts the fusion process from weight-level heuristics to feature-conditional decisions by leveraging posterior knowledge from fine-tuned features and objective metrics. Specifically, KL divergence is employed to compare feature distributions, enabling dynamic selection of the most appropriate LoRA adjustments. Moreover, objective metrics such as CLIP and DINO scores are incorporated as posterior knowledge to provide gradient-based guidance during the denoising process, further enhancing semantic alignment and visual fidelity. Extensive experiments across multiple benchmarks demonstrate that P-LoRA consistently achieves better generation performance without the need for retraining or additional supervision. While P-LoRA shows strong adaptability, its reliance on pre-defined content and style LoRAs may limit flexibility. Future work may explore extending P-LoRA to video or 3D generation tasks and developing generalization mechanisms.

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

# Appendix for P-LoRA: Posterior Knowledge Enables Training-Free Fusion of Subject and Style LoRAs

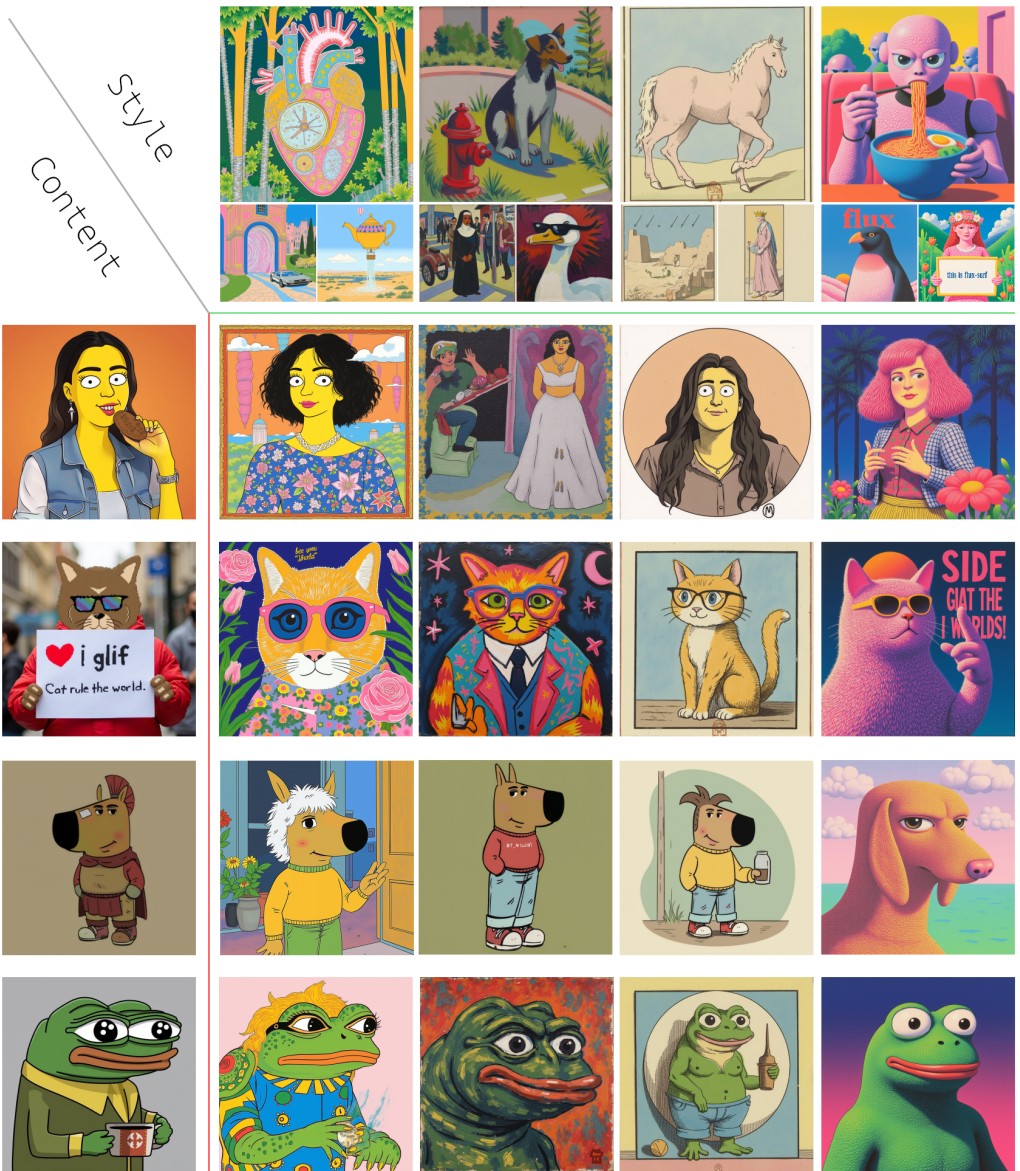

Figure 6: Additional results generated using FLUX. Each image corresponds to the object label indicated above and the style reference on the left. The results demonstrate the effects of applying different LoRA modules through our proposed method.

## A    ADDITIONAL EXPERIMENTAL RESULTS BASED ON FLUX

As discussed in the main experiments section, to more comprehensively illustrate the superior performance and generalization capability of our proposed method built upon the FLUX framework, we further conduct extensive qualitative evaluations using publicly available, well-trained LoRA (Low-Rank Adaptation) weights shared by the community on HuggingFace. Specifically, we selected a diverse set of LoRA weights corresponding to various object categories and artistic styles to systematically evaluate our model's ability to integrate and synthesize complex cross-domain

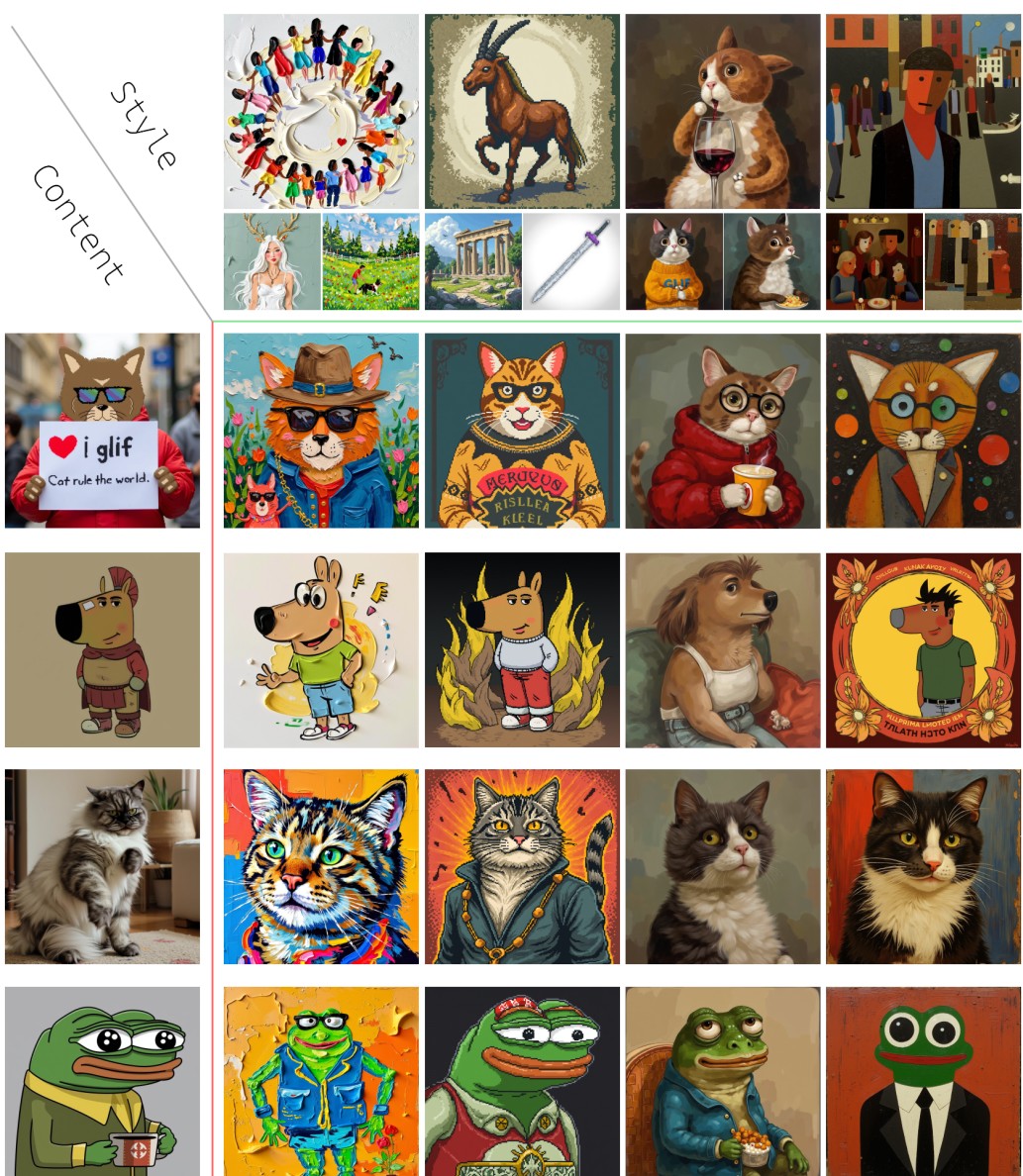

Figure 7: Additional results generated using FLUX. Each image corresponds to the object label indicated above and the style reference on the left. The results demonstrate the effects of applying different LoRA modules through our proposed method.

representations. The resulting fused images, presented in Figure 6 and Figure 7, showcase a wide range of combinations where object semantics and stylistic attributes are jointly encoded and rendered through our method.

Our approach incorporates these LoRA weights by disentangling and recombining object- and style-specific latent representations in a manner that leverages posterior knowledge extracted from both fine-tuned features and downstream objective metrics. This enables the model to align and synthesize visual content in a controlled yet flexible fashion. The generated samples exhibit not only strong fidelity to the semantic structure of the target object but also high consistency with the desired style, demonstrating the model's ability to preserve critical attributes from both input domains. Furthermore, the seamless integration of appearance and content substantiates the robustness of our framework

| Method | ViT Content ↑ | ViT Style ↑ | BLIP-2 Content ↑ | BLIP-2 Style ↑ |
|---|---|---|---|---|
| **ZipLoRA** Shah et al. (2024) | 23.2% | 18.0% | 48.2% | 46.0% |
| **B-LoRA** Frenkel et al. (2024) | 29.6% | 19.9% | 42.0% | 54.3% |
| **K-LoRA** Ouyang et al. (2025) | 32.8% | 20.7% | 49.4% | 51.1% |
| **P-LoRA (ours)** | 33.7% | 21.8% | 49.5% | 51.4% |

Table 6: Comparison of alignment results with additional evaluation metrics.

in handling varied and unseen combinations, emphasizing its potential applicability in real-world generation tasks that demand stylistic generalization and compositional creativity. Overall, these visual results provide compelling evidence of the effectiveness of our method in producing coherent, high-quality outputs across a broad spectrum of challenging scenarios.

## B  EVALUATION WITH ADDITIONAL METRICS

In addition to conventional measures such as Style Similarity, CLIP Score, and DINO Score, we further adopt ViT-based Dosovitskiy et al. (2021) and BLIP-2-based Li et al. (2023) content and style scores to provide a more comprehensive evaluation of P-LoRA. As shown in Table 6, P-LoRA achieves state-of-the-art results in three out of four metrics. A closer look reveals that different baselines exhibit complementary strengths—ZipLoRA favors BLIP-2 content alignment, while B-LoRA excels in BLIP-2 style consistency. K-LoRA maintains relatively balanced performance across metrics. In contrast, P-LoRA not only improves both ViT-based and BLIP-2-based scores, but also demonstrates a better balance between content fidelity and style preservation. We attribute this advantage to the proposed soft, inference-time metric guidance, which adaptively calibrates the generation process to maintain semantic and stylistic coherence without overfitting to a single representation space. These results suggest that P-LoRA is more robust across heterogeneous evaluation perspectives, highlighting its potential for broader generalization to multimodal generation scenarios.

## C  SETUP DETAILS FOR USER STUDY AND MLLM EVALUATIONS

Following the evaluation protocol of K-LoRA Ouyang et al. (2025), we conducted a user study where participants were presented with a **reference subject image**, a **reference style image**, and two anonymized outputs—one generated by **P-LoRA** and the other by a randomly selected baseline (ZipLoRA Shah et al. (2024), B-LoRA Frenkel et al. (2024), or K-LoRA Ouyang et al. (2025)). To mitigate presentation bias, the order of the two outputs was randomized across trials. Participants were asked: *"Which image better reflects the given artistic style while preserving the subject identity?"* We collected a total of 1,290 responses from 43 participants, with each participant evaluating a unique set of 30 trials.

Beyond human evaluation, we further adopt **GPT-4o** and **Qwen2.5-VL** as multimodal large language model (MLLM) judges to assess perceptual alignment. For each trial, the prompt included the content image, the style reference, and four anonymized outputs from ZipLoRA, B-LoRA, K-LoRA, and P-LoRA (randomized order). The LLMs were instructed to select the image that best balances style fidelity with subject preservation. To ensure robustness, we randomly sampled **100 subject–style pairs**, and each pair was evaluated in three independent runs. The final score was determined via majority voting across runs, following standard practice in recent MLLM-based evaluation studies. Importantly, while the user study captures subjective human preference, the MLLM-based evaluation provides scalable and reproducible judgments, making the two evaluations complementary and mutually reinforcing.

## LLM USAGE STATEMENT

Large language models (LLMs) were used solely as general-purpose assist tools for language editing, grammar checking, and improving clarity of presentation. They were not involved in research ideation, model design, experimental execution, or result analysis. All technical contributions, experimental designs, and findings are original work by the authors.

