# OpenReview forum: "P-LoRA: Posterior Knowledge Enables Training-Free Fusion of Subject and Style LoRAs"
_ICLR.cc/2026/Conference — ICLR 2026 Conference Withdrawn Submission_

### Official Review · Reviewer_WzzB · 2025-10-16

**Soundness:** 2
**Presentation:** 3
**Contribution:** 2
**Rating:** 4
**Confidence:** 5

**Summary:**

This paper introduces P-LoRA, a training-free method for fusing subject and style LoRAs in diffusion-based image generation. Unlike prior fusion techniques that rely on weight-level heuristics (e.g., ZipLoRA, B-LoRA, K-LoRA), P-LoRA leverages posterior knowledge derived from both fine-tuned features and objective metrics to guide the fusion process adaptively.

**Strengths:**

1. Moves from weight-based to feature- and metric-based LoRA fusion, grounded in a clear motivation that aligns better with LoRA’s original design philosophy.
2. Avoids retraining or fine-tuning, yet achieves superior results through posterior guidance.
3. The use of multiple evaluation settings (Stable Diffusion XL, FLUX, and public LoRAs) strengthens reproducibility and generality.

**Weaknesses:**

1. Although “training-free,” the method introduces extra computation (e.g., feature divergence calculation), with no runtime or memory analysis provided.
2. Benchmarks focus on synthetic subject-style datasets; results on more challenging or real-world domains (e.g., multiple styles or compositional generalization) would strengthen the claim of generality.
3. Relying on CLIP/DINO guidance ties the approach to specific pretrained models and could bias outputs toward their embedding spaces.
4. The paper lacks a formal justification for using KL divergence as the optimal criterion for feature fusion and for the gradient-based metric guidance step.

**Questions:**

See weakness

---

### Official Review · Reviewer_orLy · 2025-10-30

**Soundness:** 3
**Presentation:** 3
**Contribution:** 3
**Rating:** 4
**Confidence:** 2

**Summary:**

This paper introduces P-LoRA, a training-free framework for fusing subject LoRAs and style LoRAs in text-to-image diffusion models. Unlike previous LoRA fusion methods (e.g., K-LoRA, ZipLoRA, B-LoRA), which directly combine weights using heuristics or statistical properties, P-LoRA proposes to leverage posterior knowledge derived from: Posterior Feature Knowledge (PFK): Selecting, at each layer, the LoRA (subject or style) that induces larger KL divergence between original and fine-tuned features, under the assumption that greater feature divergence implies higher importance. Posterior Metrics Knowledge (PMK): Using external objective metrics (CLIP, DINO) as posterior signals to guide the denoising process via gradient-based updates. The authors claim this approach allows training-free, adaptive, and input-conditional fusion of LoRAs, achieving better content–style preservation without retraining. Experiments on Stable Diffusion XL and FLUX models show improvements over baselines (K-LoRA, ZipLoRA, B-LoRA) in CLIP and Style Similarity metrics, as well as higher human and MLLM (GPT-4o, Qwen2.5-VL) preference rates.

**Strengths:**

- The paper is well-written, logically organized, and the figures (especially Figure 2 and Figure 5) help illustrate the dynamic layer-wise fusion and guidance mechanisms.
- The mathematical derivations (Eq. 1–7) are clear enough for reproduction, despite some heuristic assumptions.

**Weaknesses:**

- The benchmarks are narrow: all experiments use subject–style combinations from DreamBooth and StyleDrop datasets. There are no experiments on compositional or multi-object scenarios (e.g., Mix-of-Show, CustomDiffusion setups).
- Conceptually, P-LoRA extends K-LoRA by replacing weight magnitude selection with feature divergence selection, which is an incremental improvement rather than a fundamental paradigm shift.
- There is no discussion of why posterior guidance based on image-level metrics (CLIP/DINO) should interact stably with denoising updates, nor any analysis of gradient stability.

**Questions:**

- How was the relevant LoRA list obtained? Were the LoRA weights sourced from open repositories, or were they retrained by the authors?

---

### Official Review · Reviewer_h8ek · 2025-10-31

**Soundness:** 3
**Presentation:** 3
**Contribution:** 2
**Rating:** 4
**Confidence:** 4

**Summary:**

The authors present P-LoRA, a novel representation-conditioned fusion mechanism for image stylization. The approach has two parts, the first relying on KL-divergence to select subject and style LoRA features and the second part that integrates metrics-based feedback into the diffusion process. The specific metrics are CLIP and DINO. The authors present a number of experiments, comparing P-LoRA to other LoRA variants, finding they outperform these variants in terms of CLIP and DINO. They also present the results of a user study and an evaluation with GPT-4o and Qwen2.5-VL. In all cases, their approach outperforms the baselines.

**Strengths:**

In terms of originality, the approach essentially combines the usage of KL-divergence, CLIP, and DINO simultaneously. While other approaches have used these measures individually, this is the first (to my knowledge) to incorporate them together.

In terms of quality, the work is of fairly high quality. I have concerns about the same metrics being used to guide the diffusion process (CLIP and DINO) as then used in the experiments (since Style Sim is also CLIP-based). However, the user study and the other LLM evaluations help address this concern.

In terms of clarity, the paper is overall well-written. However, there are still aspects that could be better clarified (see weaknesses below).

In terms of significance, I anticipate this work to have a significant impact on image stylization/style transfer communities and applications.

**Weaknesses:**

The major weaknesses, as addressed above come from the somewhat low originality (the approach is essentially a fusion of approaches from prior work), the potential quality concerns around the experiment metric and dataset choices, and lacking clarity on several points. There's not more to be said about the originality, so I'll leave that there.

For the quality, I recognize that there's no learning based on the selected metrics, but these metrics are still involved in the diffusion process and then appear again in the experiments. As such, it's follows naturally that P-LoRA would outperform the methods that do involve use both metrics. It's unclear the value this experimental result has. On a lesser note, while I recognize that these datasets are commonly used, they're relatively small, and comparisons with other, larger datasets would have been useful.

For the clarity concerns, I would have appreciated more detail on the user study methodology. There is a bit more methodological detail in the appendix but that still doesn't include information about user study population demographics or inter-rater reliability. Similarly, it's unclear whether images in the paper are selected at random or chosen by the authors. It would also be beneficial for the authors to clarify why there's a discrepancy between Table 2 (where P-LoRA soundly beats out the baselines) and Table 1 (where it's much closer).

**Questions:**

1. Did the authors try their approach on other datasets?
2. What were the demographics and inter-rater reliability for the user study?
3. Were the images picked randomly or cherry picked?
4. Can the authors explain the difference in results from Table 1 to Table 2?

---

### Official Review · Reviewer_7tAr · 2025-11-01

**Soundness:** 3
**Presentation:** 2
**Contribution:** 2
**Rating:** 4
**Confidence:** 4

**Summary:**

This paper aims to address the problem of how to combine two LoRAs in image generation tasks. Previous methods adjusted the combination by modifying the weight ratios of each LoRA within the injected layers. In contrast, this work further considers the KL divergence between the features of different LoRAs and those of the original DiT, using it as a criterion for selection. Additionally, extra modulation is applied at the model’s output stage.

**Strengths:**

1. The proposed method is simple, intuitive, and easy to reproduce.

2. The idea of using a metric based on the divergence between LoRA-generated features and the original DiT features seems reasonable.

3. The paper provides appropriate comparisons with relevant counterparts and includes both MLLM-based and human evaluation results.

**Weaknesses:**

1. The paper emphasizes that its approach is training-free, yet the metric requires fine-tuned features. Does this fine-tuning process need to be performed for every LoRA? If so, the method’s flexibility would be quite limited.

2. The experimental section lacks clarity. Do the types of LoRAs used in the evaluation also appear during the fine-tuning stage?

3. What would happen if the metric were computed directly from the LoRA output features without performing any fine-tuning?

**Questions:**

Please see the weaknesses section.

---

### Note · Authors · 2025-11-12

I have read and agree with the venue's withdrawal policy on behalf of myself and my co-authors.